# A Novel Doppler TRPG/AcT Index Improves Echocardiographic Diagnosis of Pulmonary Hypertension after Pulmonary Embolism

**DOI:** 10.3390/jcm11041072

**Published:** 2022-02-18

**Authors:** Olga Dzikowska-Diduch, Katarzyna Kurnicka, Barbara Lichodziejewska, Olga Zdończyk, Dominika Dąbrowska, Marek Roik, Szymon Pacho, Maksymilian Bielecki, Piotr Pruszczyk

**Affiliations:** 1Department of Internal Medicine & Cardiology, Medical University of Warsaw, 02-091 Warsaw, Poland; kkurnicka@wum.edu.pl (K.K.); barlicho@wp.pl (B.L.); olga.zdonczyk@gmail.com (O.Z.); dominika.dabrowska@wum.edu.pl (D.D.); mroik@wum.edu.pl (M.R.); s.pacho@tlen.pl (S.P.); piotr.pruszczyk@wum.edu.pl (P.P.); 2Department of Psychology, SWPS University of Social Sciences and Humanities, 03-815 Warsaw, Poland; max.bielecki@gmail.com

**Keywords:** echocardiography, echocardiographic probability of pulmonary hypertension, chronic thromboembolic pulmonary hypertension, pulmonary embolism

## Abstract

Background: We hypothesized that a Doppler index, the ratio of tricuspid regurgitation peak gradient (TRPG) to pulmonary ejection acceleration time (AcT), improves the assessment of the echocardiographic probability of pulmonary hypertension in the identification of CTEPH and chronic thromboembolic pulmonary disease (CTED) in symptomatic patients after PE. Doppler echocardiography is recommended as the initial imaging tool for the diagnosis of chronic thromboembolic pulmonary hypertension (CTEPH) after acute pulmonary embolism (PE). Methods: We analyzed the data from 845 consecutive PE (468 women; 61 ± 18 years) survivors who completed at least 6 months of anticoagulation therapy. Here, 555 patients (325 women; 66 ± 16 years) reporting functional impairment (FI) underwent transthoracic echocardiography. We included 506 patients (297 women; age 63.4 ± 16.6 years) in whom both AcT and TRPG were available into the current study. The presence of a minimum of intermediate echocardiographic probability of PH necessitated the diagnosis of CTEPH. Results: Echocardiography revealed a high echocardiographic probability of PH in 69 (13.6%) and intermediate echocardiographic probability in 109 (21.5%) patients. CTEPH was diagnosed in 35 (6.9%) patients and CTED in 22 (4.3%) patients. TRPG/AcT was significantly higher in the combined CTEPH + CTED group than in those with other causes of FI (0.412 (0.100–2.197) vs. 0.208 (0.026–0.115), *p* < 0.001), and the area under the receiver operating characteristic curve of the TRPG/AcT for CTEPH + CTED was 0.804 (95% confidence interval (CI): 0.731–0.876). Importantly, multiple logistic regression showed that TRPG/AcT is a significant predictor of CTEPH + CTED after considering echocardiographic probability (odds ratio = 1.51, 95% CI: 1.25–1.91, *p* < 0.001). Conditional inference trees analysis revealed that TRPG/AcT > 0.595 identified patients with CTEPH or CTED with a positive predictive value of 78.6% and negative predictive value of 92.7%. Conclusions: A Doppler index TRPG/AcT improves the assessment of symptomatic PE survivors. TRPG/AcT > 0.6 indicates a high probability of CTEPH or CTED, whereas TRPG/AcT < 0.6 allows for the safe exclusion of CTEPH + CTED in patients with a low echocardiographic probability of PH.

## 1. Introduction

Post-pulmonary embolism syndrome is defined as new or progressive dyspnea, exercise intolerance, and/or impaired functional and mental status after at least 3 months of adequate anticoagulation following acute pulmonary embolism (PE), which cannot be explained by other (pre-existing) comorbidities [1], and has been observed in up to 40–60% of PE survivors [2,3]. Various causes of post-pulmonary embolism syndrome have been reported, including chronic thromboembolic pulmonary hypertension (CTEPH), chronic thromboembolic pulmonary disease (CTED), deconditioning, and mental health problems [1,3,4]. Importantly, coexisting cardiopulmonary diseases may contribute to functional impairment (FI) after acute PE. The estimated prevalence of CTEPH in PE survivors is 2% and 5–8% in PE survivors with persistent dyspnea [5]. According to the current European Society of Cardiology (ESC) guidelines, all patients should be evaluated for signs and symptoms of reduced functional capacity and persistent dyspnea 3–6 months after an episode of acute PE [6]. Clinical screening with subsequent imaging diagnosis should be performed in PE survivors who report FI. The primary aim of the diagnostic approach to post-pulmonary embolism syndrome is to rule out CTEPH, because a diagnostic delay in CTEPH is associated with poor outcomes and higher risk of death [7,8]. The current ESC guidelines recommend Doppler echocardiography as the first diagnostic tool for suspected CTEPH [9,10]. When echocardiography shows an at least intermediate probability of pulmonary hypertension (PH), a diagnostic imaging workup for CTEPH is recommended [10]. However, although echocardiography is helpful for the assessment of PH probability, its diagnostic value in the identification of potential causes of PH is rather limited. Recent studies have shown that left ventricular (LV) diastolic dysfunction is the most common cause of functional limitation after PE [4,11]. However, differentiating between precapillary and postcapillary PH has been challenging [12,13]. Hence, there is a need to improve current clinical practice and diagnose CTEPH earlier [9]. Chronic thromboembolic obstruction of pulmonary arteries not only leads to a progressive elevation in tricuspid regurgitation peak gradient (TRPG), but also modifies the Doppler pattern of pulmonary artery ejection with marked shortening of the acceleration time (AcT) and midsystolic deceleration [13,14]. We hypothesized that evaluation of a novel Doppler index, the ratio of TRPG to pulmonary ejection AcT (TRPG/AcT) may extend the potential benefit of echocardiography by providing additional clues for CTEPH and CTED and improving diagnostic workup after acute PE.

## 2. Material and Methods

This was a prospective, single-center, follow-up, observational, cohort study of consecutive PE survivors, managed in a single reference center as part of the “PE-aWARE” registry (NCT03916302). After the acute PE event, all patients underwent standard outpatient follow-up and were anticoagulated for at least 6 months [3]. We included all consecutive patients with acute PE after excluding subjects with comorbidities that significantly limited survival or mobility (e.g., patients with advanced cancer or bed-ridden patients).

In all cases, the acute PE diagnosis was objectively confirmed using contrast-enhanced computed tomography pulmonary angiogram when thromboemboli were visualized in a segmental pulmonary artery.

### 2.1. Methods

At follow-up, all subjects underwent a structured clinical examination focused on FI compared to that in the pre-PE period (Figure 1). Routine laboratory tests were analyzed in all patients. Moreover, all patients reporting functional limitation relative to the pre-PE period underwent standardized transthoracic echocardiography, which was performed by an experienced cardiologist according to the current European Association of Cardiovascular Imaging recommendations [15]. In the current analysis, we included patients with both AcT and TRPG available. Patients who fully recovered after PE were referred for further clinical follow-up. Subjects with functional limitations underwent echocardiography for determinations of the probability of pulmonary hypertension. Echocardiographic probability of pulmonary hypertension was categorized as a low probability in patients with TRPG < 31 mmHg without other echocardiographic signs suggesting PH; intermediate probability in subjects with TRPG < 31 mmHg and at least two of following signs, RV/LV diameter ratio > 1.0, flattened interventricular septum, AcT < 105 ms, early diastolic pulmonary regurgitation velocity > 2.2 m/s, distended inferior cava > 21 mm with decreased inspiratory collapse, right atrial area (end-systole) > 18 cm^2^, diameter of pulmonary artery > 25 mm, or patients with TRPG in range 31–46 mmHg without the above-mentioned signs, while a high probability of pulmonary hypertension was diagnosed when additional signs were detected in this group. A high probability was defined when TRPG > 46 mmHg. Patients with at least an intermediate echocardiographic probability of PH according to the ESC guidelines [10] were referred for a detailed diagnostic workup for CTEPH, which included lung scintigraphy, pulmonary functional tests, cardiopulmonary exercise test, plasma biomarkers, CTPA, and right heart catheterization (RHC) with selective pulmonary angiography, when indicated. All symptomatic patients with a low echocardiographic probability of PH underwent additional individualized diagnostic workups driven by the clinical assessment to define the cause of FI. However, when it was inconclusive or when CTEPH was suspected by the managing physician, a diagnostic workup for CTEPH was also performed.

#### 2.1.1. Echocardiography

All echocardiograms were performed by an experienced cardiologist using a Philips IE33 or Epiq7C according to the predefined standardized protocol, and were focused on the echocardiographic criteria for suspected PH according to the 2015 ESC guidelines [10]. All examinations were performed by three experienced certified echocardiographers (O.D.D., B.L., and K.K.). During this study, we did not assess intra- and inter-observer variability, because we planned to assess the TRPG/AcT ratio in the everyday clinical scenario. However, we previously assessed and reported intraclass correlation coefficients for echocardiographic parameters to calculate inter- and intra-observer variability for the same echocardiographers [16]. The examinations were digitally recorded and reviewed when necessary. Patients were examined in the left lateral position. The dimensions of the right and left ventricles were measured in the parasternal long-axis view and apical four-chamber view at the level of the mitral and tricuspid valve tips in late diastole, as defined by the R wave of the continuous electrocardiogram tracing [15,17]. Measurements were averaged over three consecutive heart cycles. Tricuspid valve regurgitation was qualitatively assessed using color Doppler, and the peak gradient was calculated using the simplified Bernoulli formula with the tricuspid regurgitation flow peak velocity, as assessed using continuous wave Doppler [18]. The examination was completed by measuring the inferior vena cava during late expiration and inspiration with an assessment of its collapsibility. The right ventricle (RV) systolic pressure (RVSP) was calculated, and in the parasternal short axis view, flattening of the interventricular septum was assessed qualitatively, and the AcT of pulmonary ejection was measured in the RV outflow tract, just below the pulmonary valve. In M-mode presentation, RV function was assessed by measuring tricuspid annular peak systolic excursion. The left ventricular ejection fraction (LVEF) was calculated according to Simpson’s formula by employing a two-dimensional image of the LV chamber during systole and diastole in the four- and two-chamber apical views. The valvular morphology and function were evaluated according to the current recommendations [18,19]. Moreover, the left ventricular diastolic function was assessed, including mitral valve inflow (E wave), early diastolic velocities of the mitral annulus (e’) with tissue Doppler, and the mean E/e’ ratio was calculated, E/e’ ratio > 13, which indicated significant diastolic dysfunction [20,21]. All measurements were done at a trace speed of 75 mm/s.

#### 2.1.2. Diagnoses

CTEPH was diagnosed when the invasive right heart catheterization showed that the mean pulmonary artery pressure (mPAP) was ≥25 mmHg at rest on RHC, pulmonary wedge pressure ≤15 mmHg, and abnormal imaging findings on the V/Q scan, while CTED was diagnosed as CTEPH when mPAP < 25 mmHg [10].

Heart failure with preserved ejection fraction (HFpEF) was diagnosed when LVEF at echocardiography was ≥50%, while heart failure with reduced EF (HFrEF) was diagnosed when the EF was <40% [21]_._

Coronary artery disease was diagnosed when stenosis of the left main coronary artery was at least 50%, stenosis of at least 70% of epicardial arteries was found on angiography, positive stress scintigraphy with induced ischemia affecting at least 5% of the LV, or by documented acute coronary syndrome.

Valvular heart disease was defined according to the ESC [17] guidelines, with at least moderate severity.

Informed consent was obtained from each patient, and the study protocol conformed to the ethical guidelines of the 1975 Declaration of Helsinki, as reflected in a priori approval by the institution’s ethics committee.

#### 2.1.3. Statistical Analysis

Continuous variables are reported as medians with ranges (unless indicated otherwise). Between-group comparisons were conducted using the Mann–Whitney test because of the non-normality of the data distributions, large differences in group sizes, and the presence of outliers. The chi-squared or Fisher’s exact tests were used to compare nominal variables (depending on the distribution of the expected values).

The incremental validity of the proposed novel TRPG/AcT index was assessed using multivariate logistic regression. Receiver operating characteristic (ROC) analysis was used to determine the area under the curve (AUC) for predicting CTEPH and CTEPH + CTED.

Classification algorithms integrating the information obtained from EACVI recommendations with the TRPG/AcT index were estimated using conditional inference trees (with 10-fold cross validation), as implemented in the party R package, with risk levels and TRPG/AcT as predictors. *p*-values below 0.05 were considered significant for all tests and in the process of inference tree partitioning [22].

All analyses were conducted using R environment, version 4.0.5 [23].

## 3. Results

In the current study, we included 845 consecutive PE survivors (468 women; age, 61 ± 18 years) who were followed up and managed at our outpatient clinic; 65% (555/845) reported significant FI relative to the period before PE. In the current study analysis, we included 506 patients (297 women; age 63.4 ± 16.6 years) in which all echocardiographic parameters, especially AcT and TRPG, were available. The presence of a minimum of intermediate echocardiographic probability of PH necessitated the diagnostic workup for CTEPH, while patients with a low echocardiographic probability underwent individualized diagnostic workup for other causes of FI. Finally, CTEPH was diagnosed in 35/506 patients (6.9%) and CTED in 22/506 patients (4.3%) (Figure 1 and Table 1). Patients reporting FI after PE were compared to those who fully recovered were older, predominantly female, had unprovoked PE, and presented with RV dysfunction at the time of diagnosis (RV/LV > 1 on echocardiography or CTPA), while there were no differences in the type of anticoagulation (Table 2).

Among the other symptomatic patients, the most common cause of FI was HFpEF in 213 (42%) patients. Among the remaining patients with FI, major causes included CAD (60/506, 12%, coronary angiography was performed in 54 patients and scintigraphy in 6 patients), chronic pulmonary disease (44/506, 8.7%), HFrEF (32/506, 6.3%), and VHD (31/506, 6.1%). Other causes of reduced exercise tolerance included morbid obesity, neoplasm, psychiatric disorders, rheumatoid disease, and anemia (Figure 1).

### 3.1. Echocardiographic Findings

Echocardiography revealed a high echocardiographic probability of PH in 69 (13.6%) and intermediate echocardiographic probability of PH in 109 (21.5%) patients of the 506 studied subjects. Transthoracic echocardiography performed at follow-up showed signs of HFpEF in 213 patients and HFrEF in 32 patients. At least moderate aortic stenosis was newly diagnosed in 21 patients, and significant mitral valvular disease was discovered in five patients (four with mitral regurgitation and one with mitral stenosis with a mitral valve area of 1.4 cm^2^). AcT, TRPG, and RVSP differed significantly between the CTED+CTEPH group and other symptomatic patients; however, diastolic function parameters were not significantly different (Appendix A).

### 3.2. High Probability of Pulmonary Hypertension on Echocardiography

Echocardiography revealed a high probability of PH in 13.6% (69/506) of patients reporting FI (Table 3). All of them underwent diagnostic workup for CTEPH, which was the final diagnosis in 36.2% (25/69 cases); CTED was diagnosed in six other patients. Eventually, the remaining 38 patients with a high echocardiographic probability of PH were diagnosed with VHD (six patients); HFrEF (four patients); VHD and HFrEF (one patient); isolated significant diastolic LV dysfunction (13 patients); and CAD, chronic obstructive pulmonary disease (COPD), and arrhythmia (14 patients). Although most patients diagnosed with CTEPH presented with high echocardiographic probability of PH, nine patients had an intermediate echocardiographic probability of PH. Importantly, even a high echocardiographic probability of PH showed a relatively low PPV of 36.2% for CTEPH. Of note, one patient with mild CTEPH presented with a low echocardiographic probability (Table 3). Moreover, 50% of patients with CTED also had a low echocardiographic probability of PH. Table 4 shows the diagnostic value of the echocardiographic probability for CTEPH or CTEPH and CTED combined.

### 3.3. TRPG/AcT Ratio in the Diagnosis of CTEPH and CTED

TRPG/AcT was significantly higher in the CTEPH group (0.718, 0.200–2.197) than in the CTED group (0.278, 0.100–0.703, *p* < 0.001) or in patients with other diagnoses (0.208, 0.026–0.115, *p* < 0.001). Importantly, TRPG/AcT in the pooled group comprising CTEPH and CTED patients was still significantly higher than that in the other patients (0.412, 0.100–2.197, *p* < 0.001). Although TRPG/AcT tended to be higher in the CTED group when compared to the subjects with other diagnoses, there was no significant difference (*p* = 0.105) (Figure 2). Of note, for TRPG/AcT, we found significant correlations between the TRPG/AcT index and invasively obtained mPAP both in CTEPH and CTED groups (0.49, 95% CI (0.17–0.72), *p* = 0.004, and 0.56. 95% CI (0.072–0.831), *p* = 0.024 respectively). The diagnostic value of the TRPG/AcT index was evaluated using ROC curve analyses. The area under the ROC curve for CTEPH was 0.928 (95% CI: 0.885–0.971, *p* < 0.001) and 0.804 (95% CI: 0.731–0.876, *p* < 0.001) for the pooled CTEPH and CTED groups (Figure 3).

### 3.4. Incremental Validity of TRPG/AcT Index for CTEPH and CTED Diagnosis

The incremental validity of the novel index was verified using multiple logistic regression, which showed that TRPG/AcT is a significant predictor of both CTEPH and CTEPH + CTED, even if the echocardiographic stratification of PH probability was included in the model. The odds ratio of TRPG/AcT for CTEPH and CTEPH + CTED diagnosis in multiple logistic regression was 1.89 (95% CI: 1.45–2.66, *p* < 0.001) and 1.51 (95% CI: 1.25–1.91, *p* < 0.001), respectively. Detailed results of the regression models are presented in Table 5.

The AUC for TRPG/AcT was also compared with the AUCs obtained for TRPG and AcT in predicting both CTEPH + CTED and CTEPH. The novel index performed significantly better than AcT in predicting CTEPH + CTED (0.804 vs. 0.714, *p* = 0.003) and CTEPH (0.928 vs. 0.858, *p* = 0.009). Direct comparisons of ROC curves for TRPG/AcT and TRPG did not prove to be significant (0.804 vs. 0.798, for CTEPH + CTED) and (0.928 vs. 0.916 for CTEPH).

Importantly, models incorporating both the ESC echocardiographic probability of PH and TRPG/AcT were better than the regression including ESC stratification alone or even when TRPG and AcT as individual parameters were used in the model for the presence of CTEPH and CTEPH + CTED. Thus, incorporation into the proposed diagnostic algorithm of both the echocardiographic probability of PH and TRPG/AcT improved the diagnostic value echocardiography for CTEPH or CTED, which was further confirmed by the analysis based on conditional inference trees reported below.

### 3.5. Integration of ESC Stratification and TRPG/AcT Index in Predicting CTEPH and CTED

Classification algorithms integrating the ESC echocardiographic probability of pulmonary hypertension with the TRPG/AcT index were built using conditional inference trees. The algorithm was based on two predictors: ESC-based echocardiographic probability of PH and TRPG/AcT. The analyses were performed for two dependent groups: CTEPH and CTEPH pooled with CTED. Interestingly, in both cases, the obtained classification rules were identical. Figure 4 presents the final algorithm and the distribution of cases across each leaf of the tree. It is also important to underline that TRPG/AcT was the only predictor included in the models, even if TRPG and AcT were also provided in the datasets, which proves the superior informative value of the novel index in the context of both classification problems.

The conditional inference tree analysis showed that TRPG/AcT > 0.595 might be used to successfully identify CTEPH (PPV 75.0%, NPV 97.1%, specificity 98.5%, sensitivity 60.0%) and CTEPH + CTED (PPV 78.6%, NPV 92.7%, specificity 98.7%, sensitivity 38.6%). Moreover, TRPG/AcT ≤ 0.595, when combined with a low echocardiographic probability of PH, allowed us to exclude the presence of both CTEPH and CTEPH + CTED with high levels of accuracy, as indicated by NPVs of 99.7% and 96.3%, respectively. For practical clinical use, we propose TRPG/AcT ≥ 0.6 as a cutoff value in the diagnostic tree described above.

## 4. Discussion

CTEPH is a serious but curable complication of pulmonary embolism [24]. However, early diagnosis has been demonstrated to be challenging [25]. Unfortunately, untreated CTEPH can be fatal; however, if properly diagnosed in time, it may be successfully cured. It has been estimated that the majority of CTEPH diagnoses have a diagnostic delay of over 1 year [26]. Importantly, this diagnostic delay is associated with a worse prognosis, higher perioperative mortality, and inoperable disease [8]. According to the current guidelines, echocardiography should always be performed when PH is suspected [10], and assessment of signs suggesting pulmonary hypertension should be performed; in addition to tricuspid regurgitation velocity measurement, right chambers and pulmonary artery dilation, flattening of the interventricular septum, and inferior cava-decreased inspiratory collapse should be assessed [27]. The combination of these criteria allows for the estimation of echocardiographic PH probability. CTED significantly limits functional status and may lead to CTEPH. Importantly, RHC is required to confirm the diagnosis of PAH and CTEPH; however, noninvasive parameters able to predict CTEPH and CTED after acute PE would be extremely useful [24].

A meta-analysis calculated the accuracy of echocardiography vs. RHC for PH diagnosis and found a sensitivity of 83% (95% CI, 73–90%) and specificity of 72% (95% CI, 53–85%) [28,29]. Current ESC guidelines underline the role of echocardiography in the assessment of PH probability [10]; while it has limited value in the diagnosis of specific causes of PH, in some cases, echocardiography can reveal alternative causes of FI, such as LV systolic or diastolic dysfunction or significant valvular lesions [19,20].

D’Alto et al. designed an echocardiographic score for diagnosing precapillary vs. postcapillary pulmonary hypertension [12]. This score has a maximum of seven points (2 for E/e ratio ≤10, 2 for a dilated non-collapsible inferior vena cava, 1 for a left ventricular eccentricity index ≥1.2, 1 for a right-to-left heart chamber dimension ratio >1, and 1 for the right ventricle forming the heart apex). The sensitivity and specificity of an echo score of at least 2 for precapillary PH were 99% and 54%, respectively (AUC, 0.85) [12]. The D’Alto score may help in differentiating between precapillary and postcapillary PH [12]. However, it is considered cumbersome by most examiners, and an accurate measurement of its various parameters is not easy.

We performed a post-hoc analysis to identify the echocardiographic parameters suitable for identifying CTEPH and CTED patients among symptomatic survivors of acute PE. Although the majority of studied patients diagnosed with CTEPH presented with a high echocardiographic PH probability, two (6%) patients had confirmed CTEPH, and 50% of patients with CTED presented with low echocardiographic probability only. These findings suggest that even low echocardiographic PH probability does not allow for the fully safe exclusion of CTED and CTEPH. A novel TRPG/AcT index was significantly higher in patients diagnosed with CTEPH and CTED than in those with other final diagnoses of FI after acute PE (*p* < 0.001). Importantly, multiple logistic regression proved that the incremental value of the TRPG/AcT index, in addition to the echocardiographic assessment of the PH probability was a significant predictor of both CTEPH and CTEPH + CTED (OR of TRPG/AcT for CTEPH and CTEPH + CTED diagnosis = 1.89, 95% CI: 1.45–2.66, *p* < 0.001) and 1.52 (95% CI: 1.25–1.92, *p* < 0.001). The conditional inference tree analysis revealed that TRPG/AcT >0.595 had a high probability of identifying patients with CTEPH (PPV = 75.0%) and CTEPH + CTED (PPV = 76.8%). Moreover, TRPG/AcT < 0.6, when combined with low echocardiographic PH probability, allowed for the exclusion of CTEPH (NPV = 99.7%) and pooled CTEPH and CTED (NPV = 96.3%).

The TRPG/AcT ratio can be easily evaluated during routine echocardiographic examinations and may help in the management of symptomatic PE survivors. For practical clinical use, we propose a TRPG/AcT ratio of 0.6. We think that a ratio >0.6 better indicates patients with a high probability of CTEPH + CTED than a high echocardiographic probability of PH according to the ESC guidelines. We suggest that this group should undergo further invasive diagnostic workup, including RHC and pulmonary angiography. PE survivors reporting FI and presenting with TRPG/AcT < 0.6 and an intermediate high echocardiographic PH probability should undergo a noninvasive diagnostic workup. However, in symptomatic patients after PE with low echocardiographic PH probability and TRPG/AcT <0.6, alternative causes of functional impairment should be considered.

### Study Limitations

The main limitation of our study was its single-center design. Second, not all patients completed the diagnostic workup: seven patients died and 98 refused additional diagnostic procedures or were lost to further follow-up. We did not use any questionnaire to assess functional impairment, which was individually clinically evaluated by the managing cardiologist. Heart failure with preserved ejection fraction (HFpEF) was diagnosed when LVEF at echocardiography was ≥50% with signs of significant diastolic dysfunction and/or elevated NTproBNP levels. However, the NTproBNP levels were not available in all patients. Moreover, not all subjects with significant left heart pathology were diagnosed with CTEPH or CTED. It is also possible that, in some cases, chronic pulmonary thromboembolism might have also contributed to the impaired functional capacity and, in particular, the incidence of CTED may be underestimated. Moreover, in approximately 9% of symptomatic cases the TRPG/AcT index was not available; mostly due to lack of measurable velocity of tricuspid regurgitation. However, it was reported that invasively confirmed PH was present in 47% of patients with suspected PH, despite the absence of tricuspid valve regurgitation at echocardiography [30]. Importantly, it should be underlined that as TRPG is one of the criteria used for the assessment ESC echocardiographic probability of pulmonary hypertension, this limitation of affects both the echocardiographic probability of PH and the TRPG/AcT index.

Moreover, it could be of value to assess TRPG/AcT index in asymptomatic PE survivors.

## 5. Conclusions

A novel Doppler index, TRPG/AcT, improves the assessment of symptomatic PE survivors. TRPG/AcT > 0.6 indicates a high probability of CTEPH or CTED with a PPV of 78.6%. TRPG/AcT < 0.6 allows for the safe exclusion of CTEPH and CTED in patients with a low echocardiographic probability of PH with an NPV of 96.3%. However, our findings should be validated in a prospective study.

## Figures and Tables

**Figure 1 jcm-11-01072-f001:**
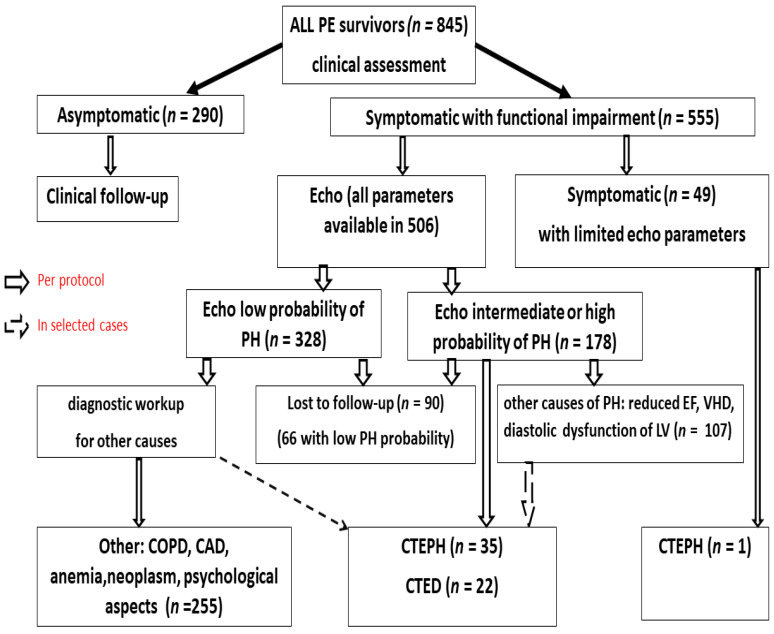
Follow up study of PE survivors. PE—pulmonary embolism, PH—pulmonary hypertension, CTEPH—chronic thromboembolism pulmonary hypertension, CTED—chronic thrombo-embolic disease, EF—ejection fraction, LV—left ventricular, COPD—chronic obstructive pulmonary disease, CAD—coronary artery disease, VHD—valvular heart disease.

**Figure 2 jcm-11-01072-f002:**
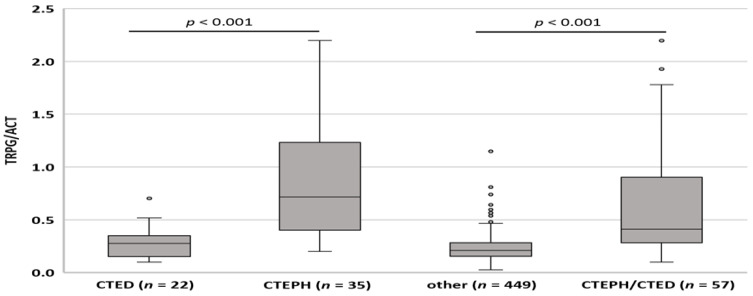
Distribution of TRPG/AcT parameter across groups. CTEPH—chronic thromboembolic pulmonary hypertension; CTED—chronic thromboembolic disease, TRPG/AcT tricuspid regurgitation peak gradient/pulmonary ejection acceleration time.

**Figure 3 jcm-11-01072-f003:**
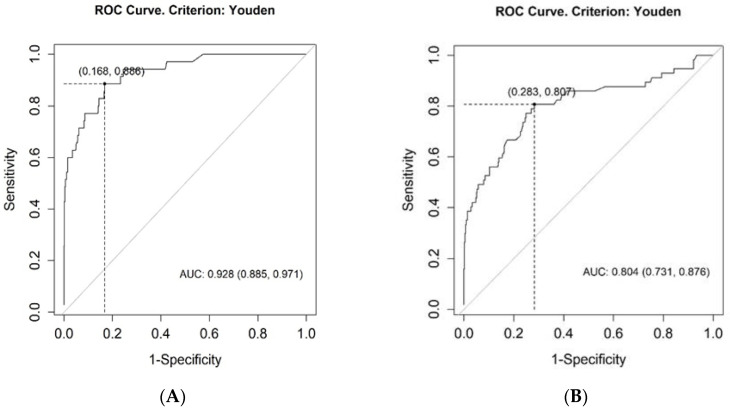
ROC Curves for the TRPG/AcT for the diagnosis of CTEPH (**A**) and for the diagnosis of pooled CTEPH and CTED groups (**B**). AUC—area under the roc curve; CTEPH—chronic thromboembolic pulmonary hypertension; CTED—chronic thromboembolic disease; ROC—receiver operating characteristic; TRPG/AcT tricuspid regurgitation peak gradient/pulmonary ejection acceleration time.

**Figure 4 jcm-11-01072-f004:**
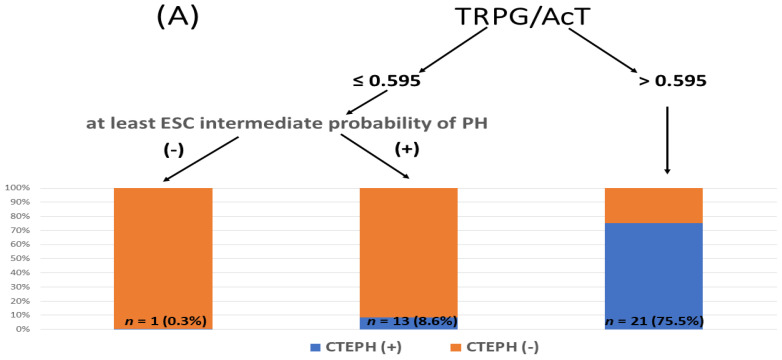
Echocardiographic diagnostic algorithm of CTEPH and CTED in pulmonary embolism survivors with functional impairment: (**A**) for CTEPH, (**B**) for combined CTEPH with CTED. AcT—pulmonary ejection acceleration time, CTED—chronic thromboembolic disease, CTEPH—chronic thromboembolic pulmonary hypertension, ESC—European Society of Cardiology, TRPG—tricuspid regurgitation peak gradient, PH—pulmonary hypertension.

**Table 1 jcm-11-01072-t001:** Right heart catheterization data of patients diagnosed with CTEPH or CTED.

	sPAP mmHg	dPAP mmHg	mPAP mmHg	PAOP mmHg	RAP mmHg	PVR Wood Unit
CTEPH (*n* = 36)	69 ± 21.8	24 ± 8	41 ± 12	9.3 ± 3.5	7.1 ± 3.3	6.4 ± 2.9
CTED (*n* = 16)	32.75 ± 7.9	9.4 ± 3.7	19.6 ± 4.6	8.75 ± 1.6	4.2 ± 1.7	2.4 ± 1.7

CTEPH—chronic thromboembolic pulmonary hypertension; CTED—chronic thromboembolic disease, sPAP—systolic pulmonary artery pressure, dPAP—diastolic pulmonary artery pressure, mPAP—mean pulmonary artery pressure, PAOP—pulmonary artery occlusion pressure, RAP—right atrial pressure, PVR—pulmonary vascular resistance.

**Table 2 jcm-11-01072-t002:** Characteristic of pulmonary embolism survivors with and without functional impairment.

Parameter	PE Survivors with FI *n* = 506	PE Survivors without FI *n* = 290	*p*
Female/Male	297/209	143/147	0.012
Age (years)	64 ± 14	49 ± 17	<0.001
RVD at PE diagnosis	300 (59%)	93 (32%)	<0.001
Unprovoked PE (*n*, %)	363 (72%)	150 (52%)	<0.001
Anticoagulant treatment at follow-up visit			
VKA	220 (44%)	136 (47%)	
DOAC	246 (49%)	136 (47%)	NS
LMWH	36 (7%)	18 (6%)	

RVD—right ventricular dysfunction; FI—functional impairment; PE—pulmonary embolism; VKA—vitamin K antagonists; DOAC—direct oral anticoagulant; LMWH—low-molecular weight heparin.

**Table 3 jcm-11-01072-t003:** Echocardiographic probability of pulmonary hypertension according ESC 2015 guidelines in 506 pulmonary embolism survivors with functional impairment.

Echocardiographic Probability of Pulmonary Hypertension	Pulmonary Embolism Survivors with Functional Impairment N/%	CTEPH N/%	CTED N/%
High (N)	69/13.6%	25/71.5%	6/27.3%
Intermediate (N)	109/21.5%	9/25.7%	5/22.7%
Low (N)	328/65%	1/2.8%	11/50.0%
Total:	506/100%	35/100%	22/100%

CTEPH—chronic thromboembolic pulmonary hypertension; CTED—chronic thromboembolic disease.

**Table 4 jcm-11-01072-t004:** Diagnostic value of echocardiographic probability of pulmonary embolism in the diagnosis of CTEPH and diagnosis of CTEPH and CTED (combined).

	**CTEPH**			
	**Sensitivity**	**Specificity**	**PPV**	**NPV**
High PH echo probability	71.4%	90.7%	19%	99.7%
Intermediate or high PH echo probability	97.1%	69.2%	19%	99.7%
	**CTEPH + CTED**			
	**Sensitivity**	**Specificity**	**PPV**	**NPV**
High PH echo probability	54.4%	91.5%	44.9%	94.1%
Intermediate or high PH echo probability	78.9%	70.1%	25.1%	96.3%

PH—pulmonary hypertension; CTEPH—chronic thromboembolic pulmonary hypertension; CTED—chronic thromboembolic disease; PPV—positive predictive value; NPV—negative predictive value.

**Table 5 jcm-11-01072-t005:** Regression models predicting CTEPH and pooled CTEPH and CTED based on the echocardiographic probability of pulmonary hypertension stratification and TRPG/AcT index.

		CTEPH			CTEPH + CTED	
Predictor	OR	95% CI	*p*	OR	95% CI	*p*
ESC risk (intermediate)	14.57	2.59–273.57	0.013	2.41	1.04–5.67	0.040
ESC risk (high)	10.88	1.32–232.84	0.047	3.55	1.13–10.69	0.026
TRPG/AcT (0.1 increase)	1.89	1.45–2.66	<0.001	1.51	1.25–1.91	<0.001
R^2^ Tjur		0.470		0.298		

AcT—pulmonary ejection acceleration time; CTED—chronic thromboembolic disease; CTEPH—chronic thromboembolic pulmonary hypertension; ESC—European Society of Cardiology; TRPG—tricuspid regurgitation peak gradient.

## Data Availability

Not applicable.

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
