# Peer review of "A Novel Doppler TRPG/AcT Index Improves Echocardiographic Diagnosis of Pulmonary Hypertension after Pulmonary Embolism"

_jcm, 2022, doi:10.3390/jcm11041072_

Round 1
Reviewer 1 Report
The research demonstrated interesting novel index for risk stratification after pulmonary embolism.
It is necessary to do comparison of ROC curves and even better NET reclassification index to further give strenghts to the main results.
Author Response
Dear Editor,
Thank you for reviewing our manuscript by O.Dzikowska Diduch et al ” A novel Doppler TRPG / AcT index improves echocardiographic diagnosis of pulmonary hypertension after pulmonary embolism” Please find below answers to Reviewers comments. All revisions according to Reviewers comments can be traced in attached version of revised manuscript “with marked changes”.
We declare that all named authors read and approved revised version of the manuscript. We are prepared to make some concessions regarding the text and/or references upon the indications by the Editor. We declare no funding and no conflict of interest.
With best regards
Olga Dzikowska-Diduch
Reviewer 1
Comment It is necessary to do comparison of ROC curves and even better NET reclassification index to further give strenghts to the main results.
Reply: As our final goal was to build a classification algorithm based on several available parameters, the most important proof of superior validity of TRPG/AcT index lies in the results of conditional inference tree analyses. These analyses showed that TRPG/AcT is preferable in building classification algorithms for predicting both CTEPH and CTEPH pooled with CTED. A relevant comment was added to the manuscript in line 247-251. The results of direct comparisons of of ROC curves for single parameters indicated superiority of TRPG/AcT to AcT (and comparable prediction to TRPG alone). This is an important information and we have added the details of these comparisons to the results section 3.4. Still, it does not invalidate our conclusions provided by logistic regression (incremental validity, when compared with EPPH stratification) and – most importantly – results of CART analyses showing that TRPG/AcT is preferred in building more complex algorithms - at least within our dataset (the need for further validation of this results is addressed in the limitations).
Reviewer 2 Report
The present study demonstrated diagnostic significance of TRPG/AcT index for pulmonary hypertension. In the present study, please provide the data of right heart catheterization of enrolled patients. Otherwise, pulmonary hypertension cannot be confirmed.
Author Response
Dear Editor,
Thank you for reviewing our manuscript by O.Dzikowska Diduch et al ” A novel Doppler TRPG / AcT index improves echocardiographic diagnosis of pulmonary hypertension after pulmonary embolism” Please find below answers to Reviewers comments. All revisions according to Reviewers comments can be traced in attached version of revised manuscript “with marked changes”.
We declare that all named authors read and approved revised version of the manuscript. We are prepared to make some concessions regarding the text and/or references upon the indications by the Editor. We declare no funding and no conflict of interest. We improved the manuscript following recommendations of all 4 reviewers.
Reviewer 4
Comment: The present study demonstrated diagnostic significance of TRPG/AcT index for pulmonary hypertension. In the present study, please provide the data of right heart catheterization of enrolled patients. Otherwise, pulmonary hypertension cannot be confirmed.
Reply: Pulmonary hypertension in all cases was confimed invasively with right heart catherization when mean pulmonary arterial pressure (mPAP) was at least 25 mmHg with wedge pressure below 15 mmHg. We are ready to provide all source hemodynamic data.
We clarified the section “Diagnoses” line 140:
“CTEPH was diagnosed when invasive right heart catheterization showed the mean pulmonary artery pressure (mPAP) was ≥25 mmHg at rest on RHC, pulmonary wedge pressure ≤15 mmHg, and abnormal imaging findings on the V/Q scan, while CTED was diagnosed as CTEPH however when mPAP <25 mmHg.

Reviewer 3 Report
The paper is interesting, but it is very difficult to read and the results to follow, due to the overabundant amount of abbreviations. The number of abbreviations should be reduced.
Since the purpose of this paper is to propose a new Doppler index, the exclusion of the 290 patients without FI could induce a selection bias. In fact we do not know if the calculation of such new index in those asymptomatic patients could have induce false positive and false negative results and which is the strenght of such index in that population. The suggestion is to analyze this new index also in that population, for comparison. If this is not possible this should be aknowledged as an other important limitation.
To help the reader, the echo definition of low, intermediate and high probability of PH, according to ESC guidelines, should be reported in more detail.
In the Echocardiography 2.1.1. paragraph a Figure or drawing could be helpful trying to show, explain and standardize echocardiographic measurements and how the dimensions of the right and left ventricles were measured in the parasternal long-axis view and apical four-chamber view. It is not clear why the curvature of the septum was assessed qualitatively and the eccentricity index was not calculated. It is also not clear why inferior vena cava was measured only at late expiration, and collapsability index was not calculated considering also inferior vena cava measurements at inspiration.
It should also be clarified if and how inferior vena cava measurements were used to calculate pulmonary artery systolic pressure from the Tricuspid peak regurgitant Velocity and gradient .
It is also not clear if mean pulmonary artery pressure and the other parameters, reported in lines 121-124, were measured and obtained by echocardiography (and how) or at right heart catheterization. Is so the definition of the paragraph 2.1.2 as "Diagnoses" should be changed and better specified.
It should be also reported if all echocardiographic measurements were performed by authors (how many?) in a blind manner as far as the clinical situation, and which were the inter and intraobserver reproducibility of measurements.
This is mostly important for the new proposed Doppler index, where the reproducibility of both components should be separately reported. For this specific index it should be specified at what recording velocity of the tracings measurements were done (25 or 50 or 100 mm/sec paper or trace speed). This could in fact affect the results.
In lines 128-131 it is not clear in how many patients and with what indications and flow chart coronary arteriography and stress scintigraphy were performed.
In lines 143-144 it is not clear how the incremental value of the new index was calculated. This incremental value should be probably derive from a comparison with the clinical value of the simple TPRG (the first component of the index). The comparison should be then performed between TPRG and TPRG/AcT indexes. Only after this comparison we could know if there is a real incremental value with this new index.
Concerning the results in the paragraphs 3.3, 3.4, 3.5, what it seems clear is that there is a wide overlap of values of the new index among different groups. This would suggest that this index could be of uncertain or limited value when it should be applied, for decision making, in the individual case.
About the 0.6 cut-off, while the specificity is high, the sensitivity is 60% and 38.6% (lines 245-246). This again suggests limited clinical value when this cut-off has to be applied in the individual case.
This is further confirmed in lines 287-288.
Furthermore in lines 295-297 the probabilty of identifyng patients with CTEPH and CTEPH+CTED is "only" 75% and 76.8%. From a clinical point of view this means in fact that we can miss one quarter of the patients.
Author Response
Reviewer 2
Comment The paper is interesting, but it is very difficult to read and the results to follow, due to the overabundant amount of abbreviations. The number of abbreviations should be reduced.
Reply: Following this recommendation we have reduced the number of abbreviations throughout the text
Comment Since the purpose of this paper is to propose a new Doppler index, the exclusion of the 290 patients without FI could induce a selection bias. In fact we do not know if the calculation of such new index in those asymptomatic patients could have induce false positive and false negative results and which is the strenght of such index in that population. The suggestion is to analyze this new index also in that population, for comparison. If this is not possible this should be aknowledged as an other important limitation.
Reply: We fully agree with the Reviewer, that it would be valuable analyze data also in asymptomatic patients. However, we tried to assess the diagnostic value in group which should be submitted to the diagnostic workup to establish the cause of functional impairment. According to current recommendations asymptomatic patients do not need to undergo echocardiography. We underline that all subjects were managed in our outpatient clinic, however due to limited resources detailed echocardiography was performed only in patients with functional impairment. Following comments acknowledged this issue as another important limitation (line 362).
Comment To help the reader, the echo definition of low, intermediate and high probability of PH, according to ESC guidelines, should be reported in more detail.
Reply Following this comment we have added following sentences (line 90-104):
“Echocardiographic probability of pulmonary hypertension was categorized as: low probability in patients with TRPG<31mmHg without other echocardiographic signs suggesting PH, intermediate probability in subjects with TRPG<31mmHg with at least two of following signs: RV/ LV diameter ratio >1.0, flattened interventricular septum, AcT< 105 ms, early diastolic pulmonary regurgitation velocity >2.2 m/sec; distended inferior cava >21 mm with decreased inspiratory collapse , right atrial area (end-systole) >18 cm2, dimeter of pulmonary artery >25mm; or patients with TRPG in range 31-46 mmHg without above mentioned signs, while high probability of pulmonary hypertension was diagnosed when in this group additional signs were detected. High probability was defined when TRPG > 46mmHg”
Comment In the Echocardiography 2.1.1. paragraph a Figure or drawing could be helpful trying to show, explain and standardize echocardiographic measurements and how the dimensions of the right and left ventricles were measured in the parasternal long-axis view and apical four-chamber view. It is not clear why the curvature of the septum was assessed qualitatively and the eccentricity index was not calculated. It is also not clear why inferior vena cava was measured only at late expiration, and collapsability index was not calculated considering also inferior vena cava measurements at inspiration.
Reply: As we indicated all patients reporting functional limitation relative to the pre-PE period underwent standardized transthoracic echocardiography, which was performed by an experienced cardiologist according to the current European Association of Cardiovascular Imaging. In order to keep the manuscript concise enough we propose not to add additional drawings. However, if the Reviewer would still recommend to add them we are ready follow this comment. We assessed diameter and collapsability index of vena cava inferior. We have clarified this in the text. (line 126).
Comment It should also be clarified if and how inferior vena cava measurements were used to calculate pulmonary artery systolic pressure from the tricuspid peak regurgitant Velocity and gradient
Reply. Pulmonary artery systolic pressure was not calculated, we reported tricuspid regurgitation peak systolic gradient. Please note that according to guidelines echocardiographic probability of pulmonary hypertension based on the tricuspid peak regurgitant velocity.
Comment It is also not clear if mean pulmonary artery pressure and the other parameters, reported in lines 121-124, were measured and obtained by echocardiography (and how) or at right heart catheterization. Is so the definition of the paragraph 2.1.2 as "Diagnoses" should be changed and better specified.
Reply Mean pulmonary pressure was measured during right heart catheterization, whereas at echocardiography TRPG was measured. We clarified the section “Diagnoses” line 140:
“CTEPH was diagnosed when invasive right heart catheterization showed the mean pulmonary artery pressure (mPAP) was ≥25 mmHg at rest on RHC, pulmonary wedge pressure ≤15 mmHg, and abnormal imaging findings on the V/Q scan, while CTED was diagnosed as CTEPH however when mPAP <25 mmHg10”
Comment It should be also reported if all echocardiographic measurements were performed by authors (how many?) in a blind manner as far as the clinical situation, and which were the inter and intraobserver reproducibility of measurements.
Reply: All examinations were performed by 3 experienced certified echocardiographers (ODD, BL, KK). Indeed during this study we did not asses intra and interobserver variability, because we planned to assess TRPG/AcT ratio in everyday clinical scenario. However, we previously assessed and reported intraclass correlation coefficients for echocardiographic parameters were calculated to assess inter- and intraobserver variation before for the same echocardiographers (Kurnicka K at al Echocardiographic Pattern of Acute Pulmonary Embolism: Analysis of 511 Consecutive Patients, JASE 2016): “Intraclass correlation coefficients for echocardiographic parameters were calculated to assess inter- and intraobserver variation by three echocardiographers in a sample of 40 randomly selected patients from the studied group. The intraclass correlation coefficients were calculated on mean values, with a value of 1 representing a perfect correlation. The inter- and intraobserver agreement for echocardiographic parameters was 94.2% (95% CI, 88.8–97.0) and 98.0% (95% CI, 96.4–99.1), respectively. The agreement analysis showed high interclass correlation for Doppler parameters (interobserver agreement = 97.3% [95% CI, 94.8%–98.6%], intraobserver agreement = 99.1% [95% CI, 97.7%–99.7%]), while intraclass correlation coefficients were only slightly lower for two-dimensional or M-mode data (interobserver agreement = 91.1% [95% CI, 82.8%–95.4%], intraobserver agreement = 96.9% [95% CI, 95.0%–98.4%]).” Moreover, regularly we perform a quality control of echocardiography examinations and measurements.
Comment This is mostly important for the new proposed Doppler index, where the reproducibility of both components should be separately reported. For this specific index it should be specified at what recording velocity of the tracings measurements were done (25 or 50 or 100 mm/sec paper or trace speed). This could in fact affect the results.
Reply: Thank you for this comment. Please see our reply to the previous comment. As we mentioned before we indented to assess the novel index for routine use hoping that it could be of clinical value. Since our results showed strong statistical significance it seem not likely that additional evaluation of reproducibility would affect our findings. All measurements were done at trace speed of 75 mm/s.
Comment In lines 128-131 it is not clear in how many patients and with what indications and flow chart coronary arteriography and stress scintigraphy were performed.
Reply: following this comment we have modified the paragraph presenting comorbidities and we specified the number of patients with coronary artery disease on the basis of coronary angiography (54 patients) or scintigraphy (only 6 patients). (line 190) “Among other symptomatic patients, the most common cause of FI was HFpEF in 213 (42%) patients. Among the remaining patients with FI, major causes included CAD (60/506, 12%, coronary angiography was performed in 54 patients and scintigraphy in 6 patients), chronic pulmonary disease (44/506, 8.7%), HFrEF (32/506, 6.3%), and VHD (31/506, 6.1%). Other causes of reduced exercise tolerance included morbid obesity, neoplasm, psychiatric disorders, rheumatoid disease, and anemia”
Comment In lines 143-144 it is not clear how the incremental value of the new index was calculated. This incremental value should be probably derive from a comparison with the clinical value of the simple TPRG (the first component of the index). The comparison should be then performed between TPRG and TPRG/AcT indexes. Only after this comparison we could know if there is a real incremental value with this new index.
Reply As our final goal was to build a classification algorithm based on several available parameters, the most important proof of superior validity of TRPG/AcT index lies in the results of conditional inference tree analyses. These analyses showed that TRPG/AcT is preferable in building classification algorithms for predicting both CTEPH and CTEPH pooled with CTED. A relevant comment was added to the manuscript in line 247. The results of direct comparisons of of ROC curves for single parameters indicated superiority of TRPG/AcT to AcT (and comparable prediction to TRPG alone). This is an important information and we have added the details of these comparisons to the results section 3.4. Still, it does not invalidate our conclusions provided by logistic regression (incremental validity, when compared with echocardiographic probability of pulmonary hypertension stratification) and – most importantly – results of CART analyses showing that TRPG/AcT is preferred in building more complex algorithms - at least within our dataset (the need for further validation of this results is addressed in the limitations).
Comment Concerning the results in the paragraphs 3.3, 3.4, 3.5, what it seems clear is that there is a wide overlap of values of the new index among different groups. This would suggest that this index could be of uncertain or limited value when it should be applied, for decision making, in the individual case.
Reply. Indeed the new index showed some overlap among different groups. Please note that finally conditional inference tree analysis formed three groups. The first one with TRPG/AcT >0.595 which successfully identifed CTEPH (PPV 75.0%, NPV 97.1%, specificity 98.5%, sensitivity 60.0%) and CTEPH+CTED (PPV 78.6%, NPV 92.7%, specificity 98.7%, sensitivity 38.6%). Whiile the other two combined TRPG/AcT ≤0.595 and echocardiographic probability of pulmonary hypertension. Importantly TRPG/AcT ≤0.595 when combined with low echocardiographic probability of PH allowed to exclude CTEPH and CTEPH+CTED with high accuracy, (NPVs of 99.7% and 96.3%). Whlie the third goup formed by patients with TRPG/AcT ≤0.595 and at least interemdiate echocardiographic probability of PH form a “grey zone“ and in our opinion they should be further diagnosed noninvasively.
Comment About the 0.6 cut-off, while the specificity is high, the sensitivity is 60% and 38.6% (lines 245-246). This again suggests limited clinical value when this cut-off has to be applied in the individual case.
This is further confirmed in lines 287-288.
Furthermore in lines 295-297 the probability of identifying patients with CTEPH and CTEPH+CTED is "only" 75% and 76.8%. From a clinical point of view this means in fact that we can miss one quarter of the patients.
Reply: It was not our intention to suggest that 0.6 cutoff score solves all the problems related to the diagnostic challenges posed by pulmonary hypertension after pulmonary embolism. Still, we believe that our results offer valuable insights and improvement over existing procedures. Firstly, the algorithm allows to select successfully the low-risk group (TRPG/AcT <.6 and low ESC risk) as indicated by 99.7% (CTEPH) and 96.3% (CTEPH+CTED) negative predictive value. Secondly, TRPG/AcT >0.6 allows to identify high-risk group (with PPV of 75% and 76.8) where more invasive diagnostic and treatment procedures would be justified. By no means we suggest the results below the cutoff should be mechanically interpreted as excluding the risk of PH. This issue is covered in more detail in the discussion section (line 341).
Reviewer 4 Report
Authors studied large cohort of pts who survived pulmonary embolism. The study is prospective, single center, observational and analyzes data acquired during follow up which is part of ambulatory care after acute episode of the disease. Aim of the study was to evaluate a novel Doppler index – TRPG/AcT as additional parameter while assessing probability of CTEPH and CTED. The research is clinically relevant because of available treatment modifications in case of proven CTEPH/CTED
The paper is neatly written. Scheme of the study is consistent with its aim, discussion summarizes main findings of the study and puts them in the light of current knowledge. As a reviewer let me share some minor remarks:
1. Functional impairment may be interpreted in many ways, is quite general description of subjective patient’s status. Was it defined on the ground of patients self-assessment or was somehow evaluated (questionnaire quality of life, scores…). Were there any identified more prevalent or characteristic forms of impairments ?
2. 42% symptomatic pts were described to have HFpEF. Did they have only echocardiographic evidence of diastolic impairment or they had HFpEF defined as clinical syndrome according to ESC 2021 definition? I did not find data related to blood concentration of natriuretic peptides. If criteria for HFpEF were not fulfilled or were in some patients – this comment/distinction should be made
Section dedicated to the limitations of the study defines some drawback however they do not diminish high quality of the paper. It is intriguing whether TRPG/AcT will turn out as valuable adjunct to echo evaluation of post PE patients as in the present manuscript
Author Response
Comment 1. Functional impairment may be interpreted in many ways, is quite general description of subjective patient’s status. Was it defined on the ground of patients self-assessment or was somehow evaluated (questionnaire quality of life, scores…).
Reply: All patients were followed in our outpatient clinic. During follow-up they underwent clinical evaluation, by an experienced cardiologist. Indeed, we did not use any dedicated questionnaire and the assessment was performed with clinical Gastalt method. All patients were asked to assess their functional status compared to the period before PE episode. All ptients reporting functional limitation relative to the pre-PE period underwent standardized transthoracic echocardiography. Following this comment we discuss this issue in the section “study limitations” (line 346).
“The main limitation of our study was its single-center design. Second, not all patients completed the diagnostic workup: seven patients died and 98 refused additional diagnostic procedures or were lost to further follow-up. We did not use any questionnaire to assess functional impairment, which was individually clinically evaluated by the managing cardiologist. Heart failure with preserved ejection fraction (HFpEF) was diagnosed when LVEF at echocardiography was ≥50% with signs of significant diastolic dysfunction and/or elevated NTproBNP levels. However, NTproBNP levels were not available in all patients. Moreover, not all subjects with significant left heart pathology were diagnosed with CTEPH or CTED. It is also possible that, in some cases, chronic pulmonary”… .
Comment 2. 42% symptomatic pts were described to have HFpEF. Did they have only echocardiographic evidence of diastolic impairment or they had HFpEF defined as clinical syndrome according to ESC 2021 definition? I did not find data related to blood concentration of natriuretic peptides. If criteria for HFpEF were not fulfilled or were in some patients – this comment/distinction should be made
Section dedicated to the limitations of the study defines some drawback however they do not diminish high quality of the paper. It is intriguing whether TRPG/AcT will turn out as valuable adjunct to echo evaluation of post PE patients as in the present manuscript
Reply: Thank you for this comment. Indeed 42% symptomatic pts were described to have HFpEF, which was diagnosed not only clinical but also when echocardiography showed preserved LVEF with E/e’ ratio exceeding 13 and/or with elevated NPproBNP plasma levels. However, NTproBNP was not always assayed. Following this comment we modified section limitations (line 350) .
“The main limitation of our study was its single-center design. Second, not all patients completed the diagnostic workup: seven patients died and 98 refused additional diagnostic procedures or were lost to further follow-up. We did not use any questionnaire to assess functional impairment, which was individually clinically evaluated by the managing cardiologist. Heart failure with preserved ejection fraction (HFpEF) was diagnosed when LVEF at echocardiography was ≥50% with signs of significant diastolic dysfunction and/or elevated NTproBNP levels. However, NTproBNP levels were not available in all patients. Moreover, not all subjects with significant left heart pathology were diagnosed with CTEPH or CTED. It is also possible that, in some cases, chronic pulmonary…”.
Round 2
Reviewer 2 Report
I understand the definition of pulmonary hypertension in the present study. I suggested to provide the hemodynamic data and then the relationship between hemodynamic data and TRPG/AcT index.
Author Response
Reviewer 2:
Comment: I understand the definition of pulmonary hypertension in the present study. I suggested to provide the hemodynamic data and then the relationship between hemodynamic data and TRPG/AcT index.
The Authors made several required changes.
Reply:
Thank you for this comment. Following this recommendation we have added additional table 4. Right heart catheterization data of patients diagnosed with CTEPH or CTED (mean followed by standard deviation)
|
|
sPAP mmHg |
dPAP mmHg |
mPAP mmHg |
PAOP mmHg |
RAP mmHg |
PVR Wood unit |
|
CTEPH (n=36) |
69 ±21,8 |
24 ±8 |
41 ±12 |
9,3 ±3,5 |
7,1 ±3,3 |
6,4 ±2,9 |
|
CTED (n=16) |
32,75 ±7,9 |
9,4 ±3,7 |
19,6 ±4,6 |
8,75 ±1,6 |
4,2 ±1,7 |
2,4 ±1,7 |
Moreover we have added information on correlation between TRPG/AcT index and invasively obtained mPAP both in CTEPH and CTED. “Of note TRPG/AcT we found significant correlations between TRPG/AcT index and invasively obtained mPAP both in CTEPH and CTED groups ( 0.49, 95% CI (0.17-0.72), p=0.004, and 0.56 95%CI(0.007-0.083, p = 0.024 respectively) (line 245-247).

Reviewer 3 Report
The Authors made several required changes.
Minor spelling ant typing check is still needed all throughout the paper: as examples see line 95 ("with" is repeated twice) ; 102 ("diameter" instead of "dimeter"); and lines 367-370.
In the methods for Echocardiography I would report, as suggested by the Authors, the phrase: "All examinations were performed by three experienced certified echocardiographers (ODD, BL, KK). Indeed during this study we did not asses intra and interobserver variability, because we planned to assess TRPG/AcT ratio in everyday clinical scenario. However, we previously assessed and reported intraclass correlation coefficients for echocardiographic parameters to calculate inter- and intraobserver variability for the same echocardiographers (Kurnicka K at al Echocardiographic Pattern of Acute Pulmonary Embolism: Analysis of 511 Consecutive Patients, JASE 2016): the reference should be reported in the list of References.
In The Echocardiographic methods I would also add, as suggested by the Authors: "All measurements were done at trace speed of 75 mm/s."
Author Response
Reviewer 3
Comment: Minor spelling ant typing check is still needed all throughout the paper: as examples see line 95 ("with" is repeated twice) ; 102 ("diameter" instead of "dimeter"); and lines 367-370.
Reply: Thank you for this comment. Typing errors were corrected.
Comment: In the methods for Echocardiography I would report, as suggested by the Authors, the phrase: "All examinations were performed by three experienced certified echocardiographers (ODD, BL, KK). Indeed during this study we did not asses intra and interobserver variability, because we planned to assess TRPG/AcT ratio in everyday clinical scenario. However, we previously assessed and reported intraclass correlation coefficients for echocardiographic parameters to calculate inter- and intraobserver variability for the same echocardiographers (Kurnicka K at al Echocardiographic Pattern of Acute Pulmonary Embolism: Analysis of 511 Consecutive Patients, JASE 2016): the reference should be reported in the list of References.
Reply: Following this comment we have added following sentences (line 118-123) and reference (16).
Comment: In The Echocardiographic methods I would also add, as suggested by the Authors: "All measurements were done at trace speed of 75 mm/s”.
Reply: Following this comment we have added following sentence (line 143).
